# Detuned Resonances

**Greg Colyer [1], Yuuichi Asahi [2] and Elena Tobisch [3],***

[1] Department of Mathematics, University of Exeter, Exeter EX4 4QF, UK
[2] Japan Atomic Energy Agency, Kashiwa, Chiba 277-0871, Japan
[3] Institute for Analysis, Johannes Kepler University Linz, 4040 Linz, Austria
* Correspondence: elena.tobisch@jku.at

**Abstract:** Detuned resonance, that is, resonance with some nonzero frequency mismatch, is a topic of widespread multidisciplinary interest describing many physical, mechanical, biological, and other evolutionary dispersive PDE systems. In this paper, we attempt to introduce some systematic terminology to the field, and we also point out some counter-intuitive features: for instance, that a resonant mismatch, if nonzero, cannot be arbitrarily small (in some well-defined sense); and that zero-frequency modes, which may be omitted by studying only exact resonances, should be considered. We illustrate these points with specific examples of nonlinear wave systems. Our main goal is to lay down the common language and foundations for a subsequent study of detuned resonances in various application areas.

**Keywords:** evolutionary dispersive PDE; nonlinear wave interactions; wave resonances

## 1. Introduction

Detuned resonances play an important role in many physical areas of research, from fluid mechanics to geophysics to engineering. Unlike exact resonances, which are comparatively rare and hard to find, detuned resonances are ubiquitous, which makes them very attractive for use in basic models. We list just a few examples. (i) A special form of detuned resonance is introduced in [1] as a new paradigm generalizing the wave kinetic equation approach for modelling nonlinear random wave fields in nature; these detuned resonances are used for numerical simulations with Zakharov's equation. (ii) A common mechanism based on the quasi-resonant amplification of planetary waves is proposed in [2], for the generation of persistent longitudinal planetary-scale high-amplitude patterns of the atmospheric circulation in the Northern Hemisphere midlatitudes. (iii) The study of detuned resonances in the numerical simulation of the Euler equations allows one to explain in which cases a discretization of a continuous problem, necessary for any numerical simulation, still reproduces the nonlinear energy transfer deduced theoretically for a continuous spectrum of water waves [3]. (iv) Finally, a fundamental drawback of vibration-based energy harvesters is that they typically feature a resonant mass-spring mechanical system to amplify the small source vibrations, and the limited bandwidth of the mechanical amplifier restricts the effectiveness of the energy harvester considerably. This problem can be solved by adaptive tuning of the input frequencies, i.e., taking into account detuned resonances [4]. The terms detuned and exact resonances both refer here to the interactions of dispersive waves. The fundamental role of dispersive wave propagation is emphasized by the fact that partial differential equations (PDEs) are often classified in physics as dispersive or non-dispersive (as distinct from their mathematical classification as elliptic, parabolic, or hyperbolic). A dispersive PDE may be linear or nonlinear; it is allowed to have an arbitrary order and number of variables. Its main characteristic is that the linear part of the dispersive PDE has solutions in the form of Fourier modes,

$$\varphi_{\mathbf{k}} = A_{\mathbf{k}} P_{\mathbf{k}}(x) exp[-i\omega_{\mathbf{k}} t], \ A_{\mathbf{k}} = \text{const} \,, \tag{1}$$

where $P_{\mathbf{k}}\left(\vec{x}\right)$ represents the spatial dependence labelled by wavevector $\mathbf{k}$, and the frequency $\omega_{\mathbf{k}} = \omega(\mathbf{k})$ [5]. Here, $x$ and $t$ are space and time variables, respectively, and $A_{\mathbf{k}}$ is called the wave amplitude. The class of nonlinear dispersive PDEs includes, though is not exhausted by, such notable equations as the nonlinear Schrödinger equation, the Korteweg–de Vries equation, and the Charney–Hasegawa–Mima equation. Dispersive waves are met in various physical systems appearing in fluid dynamics, astronomy, plasma physics, mechanical systems, medicine, and elsewhere. If the nonlinearity is small in some sense (that can be defined more precisely in a specific physical setting), the initial PDE may be written as

$$L(\varphi) = \varepsilon N(\varphi) \tag{2}$$

where $L$ is a linear operator, $N$ is a nonlinear operator, and $0 < \varepsilon \ll 1$ is a small parameter. In this case, resonances play the major role in the energy transport in the system; the resonance conditions in their simplest form for three-wave interactions with $P_{\mathbf{k}}(\mathbf{x}) = exp[i\mathbf{k} \cdot \mathbf{x}]$ read

$$\omega_1 + \omega_2 = \omega_3,\ \mathbf{k}_1 + \mathbf{k}_2 = \mathbf{k}_3 \tag{3}$$

with notation $\omega_j = \omega(\mathbf{k}_j)$; their solution is called a resonant triad. In the standard case of periodic boundary conditions, the wavevectors, after suitable renormalization, have integer wave numbers (components).

A most interesting property of the resonant triads, first discovered in the early 1990s [6–8], is that they form clusters (via joint modes belonging to different triads) that are independent in the sense that there is no energy transport between different clusters at the slow time scale $T = t/\varepsilon$. Moreover, most modes are not resonant and do not take part in the energy transfer over the resonant spectrum. The set of resonant triads can be computed by specially developed methods (this is a nontrivial problem of number theory) and represented by a planar graph with marked vertices called an **NR**-diagram [9]. An **NR**-diagram allows one to represent uniquely the dynamical system on $A_{\mathbf{k}}(T)$ which describes the time evolution of each cluster at the slow time scale $T$. In this context, $t$ is often referred to as the linear (or fast) time. All polynomial conservation laws can be written out explicitly for an arbitrary cluster [10]. An important example of geophysical application of this theoretical approach can be found in [11].

This clear and understandable picture will change if we wait long enough. Detuned resonances will appear with resonance conditions differing from (3):

$$\omega_1 + \omega_2 = \omega_3 + \delta,\ \mathbf{k}_1 + \mathbf{k}_2 = \mathbf{k}_3 \tag{4}$$

with nonzero resonance detuning $\delta \neq 0$. They take place at the new time scale $\tau = cT$, with some constant $c > 1$ depending on the wave system [12]. It has been demonstrated analytically that an isolated detuned triad can also be regarded as an energy conserving system (though its energy is oscillating with the frequency $\delta$) and an explicit solution for wave amplitudes can be written out in terms of Jacobian elliptic functions [13]. This means that the behavior of an isolated detuned triad is similar to that of the exact resonance triad. In the last few years, an interest has arisen in the study of clusters of detuned triads. By varying the magnitude of the detuning, one can change the form of energy transport in the system from a few isolated clusters (detuning is very small) to the case when all modes in the system are connected and no isolated cluster survives (detuning is big enough). Some questions to answer are: How to estimate the minimal detuning threshold allowing for the formation of a new cluster? How does the density of clusters depend on the magnitude of the detuning and on the size of the computation domain? How to construct new conservation laws for detuned clusters? These problems are studied in many recent publications, e.g., [14–16]. They are not of purely theoretical interest: the answers will be of importance in many physical systems. A celebrated example of a system in which detuned resonances play a major role can be found in plasma physics and in planetary atmospheric physics where so-called quasilinear models are used to solve

the Charney–Hasegawa–Mima or barotropic vorticity equation, e.g., [17–19]. The main characteristic of this approach is that (at a suitable time scale) all exact resonances can be omitted and the complete wave field evolution can be described in terms of detuned resonances of a special type. All the studies of detuned triad clusters mentioned above have one common feature: they are performed at the kinematic level and no study of cluster dynamics is included. The basic object underlying these studies is not an **NR**- diagram (which provides a one-to-one correspondence between a cluster and its dynamical system) but a topological presentation of a cluster [9]. The topological presentation of a cluster has been introduced for constructing all the isolated components of a given set of triads. It gives no information about cluster dynamics; the dynamical system describing a cluster cannot be derived from its topological presentation. This drawback is of course known to researchers in the area [14]: "In order to better understand these issues, we believe that it is important to move beyond the kinematic picture of resonance broadening and attempt to devise methods of studying these effects dynamically". A first step in this direction has been taken in [15], where the dynamics of a particular cluster formed by two triads with two joint modes is studied while investigating the case "the linear wave time scales are comparable to the time scales of nonlinear oscillations". In the pioneering work of [20], the study of dynamic equations for resonances in the famous Fermi–Pasta–Ulam problem made it possible to establish that the first nontrivial resonances correspond to six-wave interactions and thus to determine the time scale for establishing equipartition.

In this paper, we aim to present a feasible platform for studying the dynamical properties of arbitrary clusters formed by detuned triads. The notion of "a detuned triad" is chosen to be used as a shorter name instead of "a resonance with non-zero frequency mismatch" and is regarded as generic. The notions of quasi-resonance, near-resonance, approximate resonance, etc., describe particular cases of the detuned resonance; their interrelations will be described in Section 2. In Section 3, we demonstrate that detuning cannot be chosen arbitrarily, and that the notion of a characteristic detuning may be misleading. In Section 4, we demonstrate that zero-frequency modes, which usually are not regarded while studying exact resonances, should be taken into account while investigating clusters of detuned resonances. In Section 5, we demonstrate why the construction of a cluster of detuned triads is much more complicated than in the case of exact resonances, and we consider which minimal detuned clusters are important to study first. A brief discussion and a list of open questions concludes the paper.

## 2. Notion of a Detuned Triad

Any equation of mathematical physics is applicable only in some specific context: for given ranges of the parameters and variables. In particular, wave resonances may be studied for long waves with discrete spectra, so after a suitable renormalization, the wavenumbers, that is, the components of the wave vectors, can be regarded as integers. Below in this presentation, we restrict ourselves to the PDEs (2) with two space dimensions; in this case, each wavevector $\mathbf{k} = (k_x, k_y)$ can be envisioned as a node of the two-dimensional lattice with coordinates $(k_x, k_y)$ and to each node, we can prescribe a unique frequency $\omega = \omega(k_x, k_y)$. Re-normalized coordinates are usually depicted as integers $m$ and $n$. The small nonlinearity in (2) allows one to reduce the study of the complete wave evolution in the frame of a weakly nonlinear PDE to the study of only exactly resonant modes, i.e., modes with wavevectors and frequencies satisfying (3) (at the corresponding time scale, of course). This reduction can be performed by any multiscale method and as a result, a dynamical system can be deduced:

$$\frac{\mathrm{d}A_1}{\mathrm{d}T} = V_{23}^1 A_2^* A_3, \quad \frac{\mathrm{d}A_2}{\mathrm{d}T} = V_{31}^2 A_1^* A_3, \quad \frac{\mathrm{d}A_3}{\mathrm{d}T} = -V_{12}^3 A_1 A_2 \tag{5}$$

which is energy conserving, integrable, and describes the envelope evolution in the slow time $T$ of the amplitudes of each triple of resonantly interacting modes

$$\varphi_{\mathbf{k}} = A_{\mathbf{k}} \exp\{i[\mathbf{k}\cdot\mathbf{x} - \omega_{\mathbf{k}}t]\}, \quad A_{\mathbf{k}} = A_{\mathbf{k}}(T) \neq \text{const.} \tag{6}$$

It may happen that a specific system has no three-wave resonances; then, we have to check whether four-wave resonances do occur, and so on. Note that the general form of (5) is the same for all evolutionary dispersive weakly nonlinear PDEs possessing three-wave resonances: all the differences between the equations are hidden in the form of the interaction coefficients $V_{23}^1$, which are deduced from the PDE and are a function of wave numbers and parameters included in the PDE.

Any solution of (3) may thus be presented as a unique triangle on the integer lattice (shown as the bold black triangle in Figure 1), and the corresponding dynamical system has the form (5).

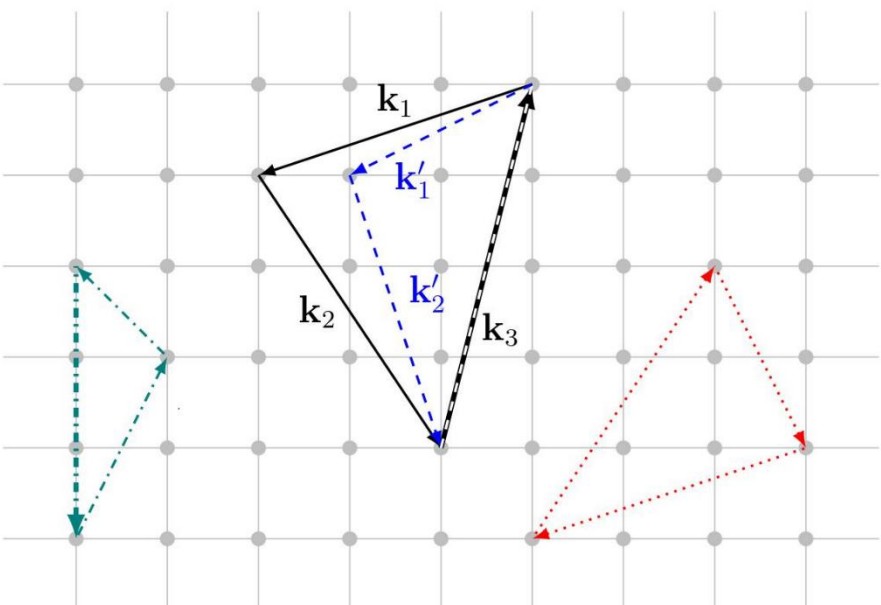

**Figure 1.** Color online. Triads on an integer lattice. Each mode has an integer length in both directions, $\mathbf{k} = (k_x, k_y) = (m, n)$. Each triad satisfies $\mathbf{k}_1 + \mathbf{k}_2 = \mathbf{k}_3$ and therefore forms a triangle (the location of the triangle on this grid has no significance). The triads shown represent a resonant triad (black, solid) and a quasi-resonant triad (blue, dashed) in its one-vicinity (also called a near-resonance); a general detuned triad (red, dashed); and a zonal triad (teal, dot-dashed), which is a (detuned) triad that includes a zonal mode (has $k_x = m = 0$ and is therefore vertical).

Now, let us see what happens when we regard a detuned resonance, satisfying the conditions (4), in the same way. Obviously, for any three nodes of the two-dimensional lattice, the condition $\mathbf{k}_1 + \mathbf{k}_2 = \mathbf{k}_3$ is satisfied so that only the condition $\omega_1 + \omega_2 = \omega_3 + \delta$ has to be verified and studied. Intuitively, it looks reasonable that among all detuned resonances, those are most important which are in some sense closest to exact resonances.

Accordingly, the first subclass of the class of detuned resonances that we consider is called quasi-resonances, and is defined by the closeness on the integer lattice. The simplest way to construct a quasi-resonance is to take two of the three nodes constituting an exact resonance (depicted as the bold black triangle) and one node neighboring with it. As a result, we get a quasi-resonance (depicted by the blue dotted lines) lying in the one-vicinity of the exact resonance. The one-vicinity of an exact three-wave resonance may consist of a few dozen quasi-resonances. Similarly, one can compute the two-vicinity, three-vicinity, etc. This type of construction (or what can be reduced to it) is used where possible ways of energy transfer within a cluster of detuned resonances were looked at. The one-vicinity of an exact resonance is sometimes called a near-resonance.

Another subclass of detuned resonances is called *approximate resonances*, defined as follows (e.g., [1]—actually used for a four-wave system therein). In some finite domain of wavevectors, the resonance mismatch should not be too big, that is, $\delta < \omega_{min} \cdot \lambda_\omega$ with $\omega_{min}$ being the minimal wave frequency in the chosen finite domain and $\lambda_\omega$ a detuning parameter, allowing one to manipulate the number of approximate triads in the domain. For some choice of a detuning parameter, the set of approximate resonances might coincide with a specific *s*-vicinity of an exact resonance, $s = 1, 2, \ldots$. This approach is then used for a statistical description of systems possessing exact resonances and many approximate triads simultaneously: the initial PDE is transformed into a simpler wave kinetic equation considering these specifically chosen detuned resonances.

Obviously, in any finite spectral domain, taking a big enough *s*-vicinity of a resonance (i.e., quasi-resonances) or, similarly, a big enough detuning parameter $\lambda_\omega$ (i.e., approximate resonances), one comes to the same result—the finite set will contain all modes as well as the resonant ones, i.e., all detuned resonances.

The main question now is how to choose, if possible, a reasonably small set of detuned resonances which plays the major role in the wavefield evolution on the corresponding time scale $\tau = cT, c > 1$, similar to exact resonances at the time scale $T = t/\varepsilon$? The magnitude of detuning $\delta$ is a relevant parameter to look at.

## 3. The Choice of Detuning

### 3.1. Detuning on a Lattice

Let us consider a finite $N \times N$ lattice, as in Figure 1 compute the value of $\omega = \omega(m, n)$, and then compute the finite list of all possible detunings for a given dispersion function $\omega$ and $N$. We get altogether $(N^2)!/3!(N^2 - 2)!$, some of them being equal to zero (exact resonances). The number of zeros depends on the number of exact resonance triads in the domain and on the form of resonance clusters. Let us denote the set of all non-zero detunings $\delta_j$ as

$$\Delta_{lattice} = \{\delta_1, \ldots, \delta_p\}; p \in \mathbb{N}. \tag{7}$$

The dynamical system describing an isolated detuned triad reads

$$\frac{dA_1}{dT} = V_{23}^1 A_2^* A_3 e^{-i\tilde{\delta}T}, \ \frac{dA_2}{dT} = V_{31}^2 A_1^* A_3 e^{-i\tilde{\delta}T}, \ \frac{dA_3}{dT} = -V_{12}^3 A_1 A_2 e^{+i\tilde{\delta}T} \tag{8}$$

with $\tilde{\delta} = \delta/\varepsilon$, and it can be turned into (5) by putting $\tilde{\delta} = 0$. Moreover, as demonstrated in [13] on page 136, in some well-defined sense, the system (8) can be regarded as an energy-conserving system where the total energy $E_1 + E_2 + E_3$ of the detuned system (8) oscillates periodically with frequency $\tilde{\delta}$, again at the specified time scale. Here, $E_1, E_2, E_3$ are energies of each distinct mode of the triad, $E_i \sim A_i^2, i = 1, 2, 3$.

The first and most important consequence of the existence of the set of allowed detunings $\Delta_{lattice}$ is the following: *the detuning cannot be made arbitrarily small*. Given a dispersion function, one can obtain a lower boundary $\delta_{min}$ for this set. For instance, if $\omega \cong m/n(n + 1)$ (spherical Rossby waves [11]), then $\delta_{min} = O(N^{-6})$. This lower estimate is "local" in that it depends on $N$. One can argue that as the size $N$ of the spectral domain tends to infinity, $\delta_{min}$ therefore tends to zero. However, this would bring us outside the applicability domain of the barotropic vorticity equation and spherical Rossby waves. The spectral domain usually studied in the problems of weather prediction is rather small, e.g., $N = 21$ and sometimes even smaller, while the characteristic length of the Rossby wave in the Earth's atmosphere is of an order of 1000 km. Such a "local" lower boundary exists for arbitrary wave systems.

For some forms of dispersion function, there exists a global lower boundary, which does not depend on any finite domain. For instance, surface water waves have a dispersion function $\omega \cong (m^2 + n^2)^{1/4}$ and four-wave exact resonances. It can be shown that if $\omega_1 \pm \omega_2 \pm \omega_3 \pm \omega_4 \neq 0$, then, $\omega_1 \pm \omega_2 \pm \omega_3 \pm \omega_4 > 1$, i.e., $\delta_{min} = 1$. More examples and explanations can be found in [9,21]. This means that the idea of making the detuning

smaller and smaller in order to get back to the case of exact resonances does not work. The question now is why look for the detuned triads in a vicinity of an exact resonance at all? Is it possible that the triads with smallest achievable detuning do not lie in the close vicinity of an exact resonance triad?

To answer this question, we need to quantify the idea of closeness, which we do here by defining a Euclidean distance $d$ between the triad $\{(m_1, n_1)(m_2, n_2)(m_3, n_3)\}$ and the triad $\{(m'_1, n'_1)(m'_2, n'_2)(m'_3, n'_3)\}$. This is given by

$$d = \min_p \sqrt{\sum_{i=1}^{3}(m_i - m'_j)^2 + (n_i - n'_j)^2} \tag{9}$$

where the indices $\{j\}$ represent a permutation $P$ of $\{i = 1, 2, 3\}$ and the permutation giving the minimum distance is taken. For each detuned triad, we can compute the distance $d$ to each resonant triad, and hence find the closest resonant triad(s) (with the least $d$) for that detuned triad. Figure 2 shows how this closeness varies with detuning for all detuned resonances in the $N = 21$ domain. Two key features are immediately apparent. First, as perhaps expected, the top-left and bottom-right corners of the figure are empty; so, the smallest detunings do not occur with the largest distances, and the largest detunings do not occur with the smallest distances. Second, however, and perhaps less expected, the rest of the figure is quite broadly filled out: in particular, the smallest detunings are not limited to the smallest distances, nor are the smallest distances limited to the smallest detunings.

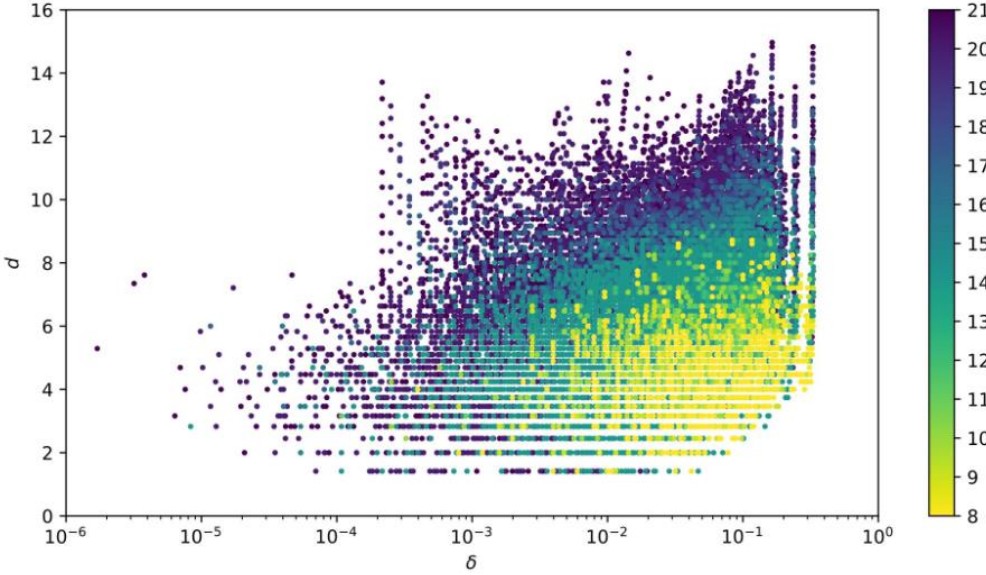

**Figure 2.** Color online. Scatter plot showing, for each detuned triad, the detuning $\delta$ versus the distance $d$ to the closest resonant triad. The color of each point indicates $n_{max}$, the largest $n$ found amongst all the six modes in the detuned triad and its closest resonant triad. The plot can be thought of as a layer of each color, with layers corresponding to low $n_{max}$, appearing in front of layers corresponding to high $n_{max}$. The number of layers is limited by the domain size $N = 21$.

One may argue that the search for quasi-resonances in some small vicinity of an exact resonance might work for a special initial distribution of the energy in the wave field—energy should be initially distributed only among exact resonant modes, and at least in some triads, high-frequency modes have to be excited—but this should be explicitly discussed as is done, e.g., in [8,22].

### 3.2. Detuning Outside the Lattice

One of the main facts established in the previous section—the allowed detuning cannot be too small—is somewhat counter-intuitive. What would prevent us from taking a very

small detuning $\delta < \delta_{min} \in \Delta_{lattice}$ and performing a numerical simulation? However, because the dynamical system is a reduction obtained from the initial PDE, some properties of the PDE may be lost; in particular, the existence of allowed and non-allowed detunings cannot be detected at the level of the dynamical system. It is easy to check numerically whether at least some "memory" about the discreteness of $\Delta_{lattice}$ survives and in what form, by performing simulations with the system (8) for different values of $\delta$ regarding it as a continuous parameter.

Before proceeding this way, let us try to understand what it would mean if the detuning does not belong to the set $\Delta_{lattice}$. A geometric interpretation of this case is shown in Figure 3 and gives us an immediate insight: as all eigenmodes of the initial PDEs are depicted as nodes of the lattice, we have to *change the initial PDE* by adding a new term, say, $D$. Accordingly, triads with allowed and forbidden detunings are obtained from different initial PDEs:

$$L(\varphi) = \varepsilon N(\varphi) \; versus \; L(\varphi) = \varepsilon N(\varphi) + D. \tag{10}$$

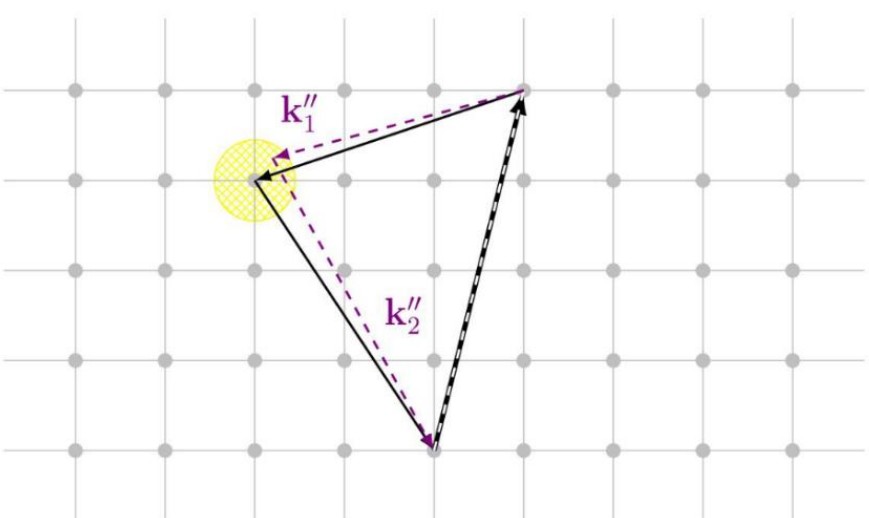

**Figure 3.** Color online. A resonant triad (black, solid) is shown, as in Figure 1, and an off-grid near-triad (violet, dashed) in its $\delta$—neighborhood (yellow disc, hatched); the two labelled modes do not lie on the lattice, i.e., do not satisfy the original system of PDE and boundary conditions.

Here $D \neq 0$ can be interpreted as a forcing or a dissipation. It follows that for $\delta \neq 0$, the wave amplitudes might become larger for the case of exact resonance described by (5): a detuned resonance in the first system may become an exact resonance in the latter and have a larger amplitude.

In order to establish whether or not this effect can be observed at the level of the dynamical system, numerical simulations have been performed with the dynamical system (8) for specific resonant triads of spherical Rossby waves, taken from [11]. More details of similar computations can be found in [23]. A typical result is shown in Figure 4.

A certain overall symmetry with respect to $\delta$ can be seen in Figure 4a—inverting the sign of $\delta$ whilst changing $\phi$. by $\pi$. However, for a fixed $\phi(0)$, the fluctuation range is not in general symmetric about $\delta = 0$, nor does it peak there, as it is demonstrated in Figure 4b.

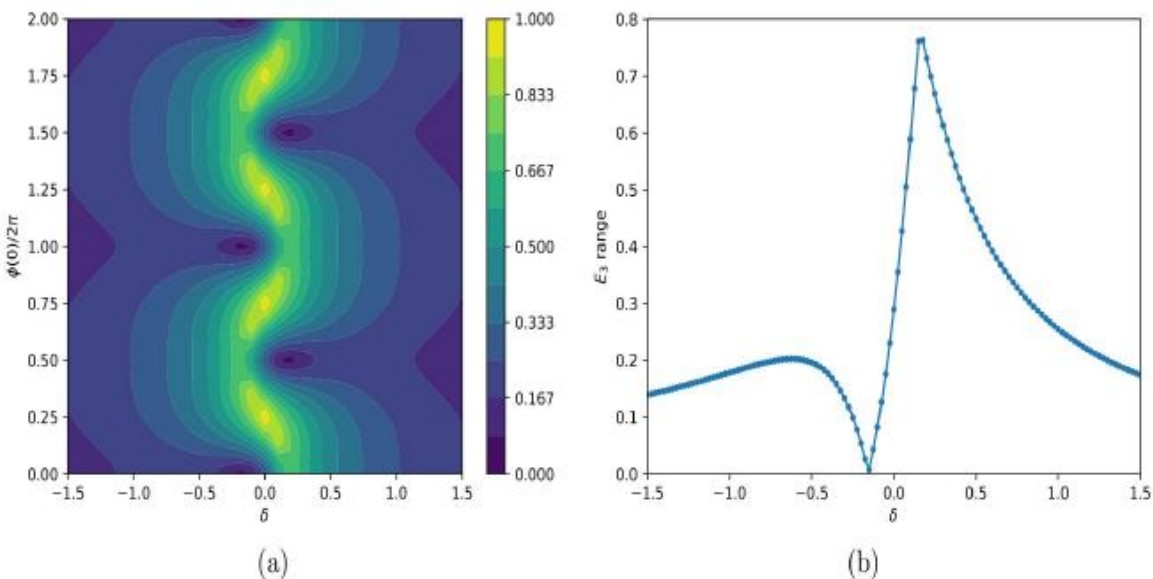

**Figure 4.** Color online. The fluctuation range of the energy $E_3$ for the triad $(4, 12)(5, 14)(9, 13)$ which is an isolated resonant train in the truncation $N = 21$, [11]: (**a**) as a function of (artificial) detuning $\delta$ and initial phase $\phi$ (at time $t = 0$; two $2\pi$ periods in $\phi(0)$ are shown); (**b**) as a function of $\delta$ at $\phi(0) = 0$. The initial energies are $E_1(0) = 0.4, E_2(0) = 0.4, E_3(0) = 0.2$, which is case (**b**) of [16].

## 4. Zero-Frequency Mode

Often, zero-frequency modes are not taken into account while studying exact resonances, because they do not satisfy the exact resonance condition. For example, for spherical Rossby waves, the dispersion function has the form $\omega(m, n) = \frac{-2m}{n \, (n+1)}$. and corresponding resonance conditions read

$$\frac{m_1}{n_1 \, (n_1 +1)} + \frac{m_2}{n_2 \, (n_2 +1)} = \frac{m_3}{n_3 \, (n_3 +1)}, \; |n_1 - n_2| < n_3 < |n_1 + n_2|,$$
$$m_j \leq n_j \; \forall \, j = 1, 2, 3,$$
$$n_1 + n_2 + n_3 \text{ is odd, } n_i \neq n_j \; \forall \, i, j = 1, 2, 3.$$

If $\omega_3 = \frac{-2m_3}{n_3 \, (n_3 +1)} = 0 \iff m_3 = 0$, then the selection rule $m_1 + m_2 = m_3$ implies $m_1 = -m_2$. However, the selection rule $n_1 \neq n_2$ then implies $\omega_1 + \omega_2 = \frac{2m_1}{n_1 \, (n_1 +1)} + \frac{2m_2}{n_2 \, (n_2 +1)} \neq 0$ and so

$$\omega_1 + \omega_2 \neq \omega_3.$$

However, zero-frequency (or zonal) modes can participate in detuned resonances (an example is shown in Figure 5). An example in the case of spherical Rossby waves is the triad between the following three $(m; n)$ modes: $(0; 2); (1; 3) \rightarrow (1; 4)$. This triad satisfies all of the kinematic selection rules for the mode numbers, just not the exact dynamical resonance condition for the $\omega$'s. Furthermore, since the three $n$'s are all different, the interaction coefficients in Equation (8) do not vanish, and the dynamical system is of the same form as for a general detuned resonance: the dependence of the dynamics on the specific triad enters only through the interaction coefficients and the detuning.

In the next section, we illustrate the dynamics of a detuned resonance, both in isolation and when coupled to another (exact) resonance (Figure 6). A particular case of the isolated zonal triad $(0; 2); (1; 3) \rightarrow (1; 4)$ will also be shown (Figure 7).

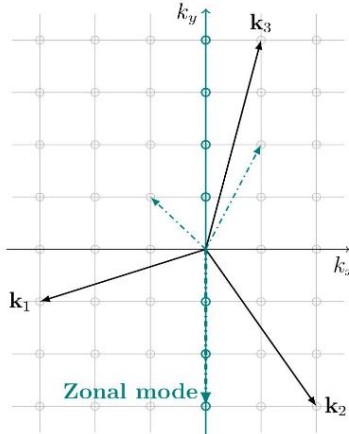

**Figure 5.** Color online. Triads on an integer lattice 2. Here, each lattice point represents a particular mode $(k_x, k_y) = (m, n)$ . The resonant triad (black, solid) and the zonal triad (teal, dot-dashed) from Figure 1 are shown. The zonal mode has $k_x = m = 0$ and therefore lies on the $k_y$ axis.

**Figure 6.** Color online. Each panel shows the time evolution of (top) the amplitude of the modes in the resonant triad; (middle) the amplitude of the modes in the detuned triad; (bottom) the energy $E$ of the resonant triad, the energy $F$ of the detuned triad and the total energy. The panels are as follows: (**a**) uncoupled with physical detuning $\delta = 0.000396825$; (**b**) coupled with physical detuning; (**c**) uncoupled with zero detuning; (**d**) coupled with zero detuning.

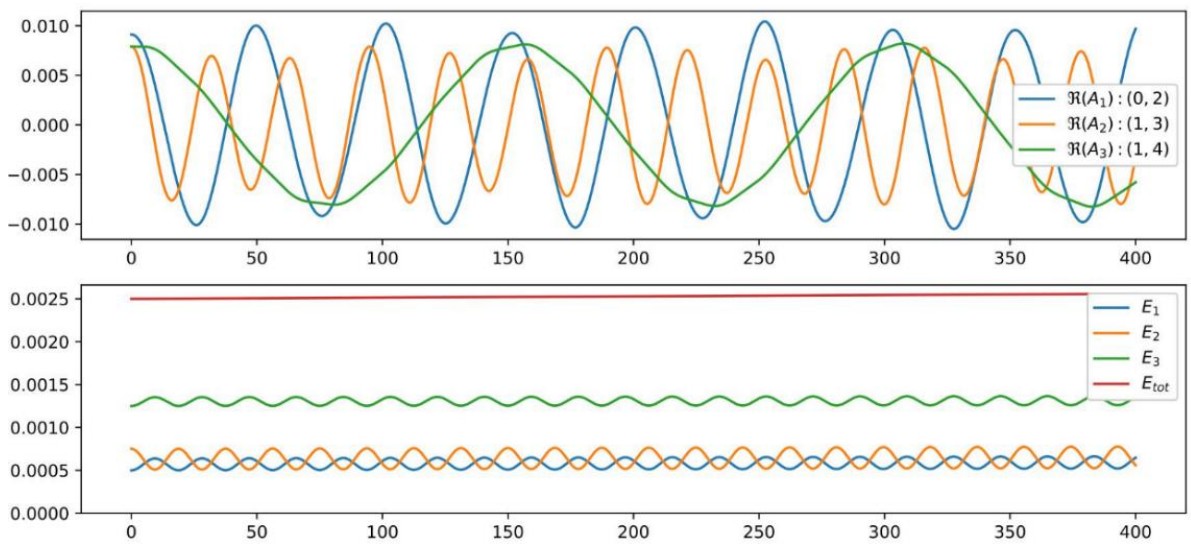

**Figure 7.** Color online. Time evolution of the amplitude (**top**) and energies (**bottom**) of the modes in the zonal triad $(0; 2)$; $(1; 3) \to (1; 4)$ with physical detuning $\delta = 0.033333333$ and $Z = 7.82$.

## 5. Clusters of Detuned Triads

The examples shown below all involve a single resonant triad and a single detuned triad. The dynamical equations are modelled on an actual example taken from the spherical Rossby system, namely the resonant ($\delta = 0$) triad (2,6), (3,8)$\to$(5,7) and the detuned ($\delta = 0.0004$) triad (1,5), (1,8) $\to$ (2,6). At the truncation $N = 8$, the former is the only resonant triad, and the latter is the detuned triad with the smallest detuning. In reality, these two triads are coupled via their common mode (2,6). In some of the simulations shown, this coupling is artificially disconnected; in others, the detuning is artificially set to zero. This is done without changing anything else about the dynamical system, so that the difference made by the coupling and the detuning may be seen in isolation.

The form of the coupling coefficients $V_{23}^1$ in the dynamical equations for three resonantly interacting spherical Rossby waves are of the form $Z \left[ n_i \left( n_i + 1 \right) - n_j \left( n_j + 1 \right) \right] / n_s \left( n_s + 1 \right)$, where $Z$ is a property of the triad and the indices $i, j, s$ are a permutation of 1,2,3. Here, for illustrative purposes, we allow ourselves to vary $Z$ for the resonant triad and $Z_1$ for the detuned triad, keeping the initial conditions fixed. As a period of energy exchange within a triad depends on the initial energy distribution within the triad and on the coefficient $Z$, this allows us to vary the relative time scales in dynamics. Figure 6 depicts characteristic dynamics of the modes' energies in four different cases: uncoupled resonant and detuned triads (physical detuning), panel (a); coupled triads (physical detuning), panel (b); uncoupled triads (artificial zero detuning), panel (c); coupled triads (artificial zero detuning), panel (d). The resonant triad has $Z = 7.82$, and the detuned triad has $Z_1 = 2Z$. This allows us to adjust the relative time scales in the uncoupled dynamics with zero detuning, as seen in the top and middle rows of Figure 6c. The slowest oscillation time scale, seen in the middle row of Figure 6a, is given by the detuning period $2\pi/\delta$. It is evident in the top row of Figure 6b that both the fast and the slow dynamics of the detuned resonance are communicated to the exact resonance when coupled.

We demonstrate in Figure 7 how the amplitudes and energies of modes in the isolated zonal triad $(0; 2)$; $(1; 3)\to(1; 4)$ evolve in time. The coefficient $Z = 7.82$ as above but the physical detuning $\delta = 0.033333333$ is much larger than for the detuned resonance in Figure 6, and hence affects much shorter time scales (seen in the varying heights of the peaks in the top panel). The mode amplitudes have been increased correspondingly so that the triad dynamics also take place on a shorter time scale (seen as the underlying oscillations in both panels).

### 6. Brief Discussion and Conclusions

Exact and detuned resonances represent a subject of widespread multidisciplinary interest describing many physical, mechanical, biological, and other systems. While the theory of exact resonances and their standard description in the form of resonance clusters, **NR**-diagrams, etc., is quite developed, the theory of approximate resonances is only in its infancy. To the best of our knowledge, this is the first attempt to introduce some systematic terminology to the field and give illustrative examples confirming the importance of understanding what resonance detuning is, why it cannot be arbitrary, what properties of a dynamic system cannot be attributed to the original PDE, etc. Both exact and detuned resonances play important roles—at different time scales—in energy transport in (weakly) nonlinear dispersive wave systems. They may be studied within at least three different mathematical frameworks: (a) the initial PDE (full evolution equation), with chosen boundary conditions; (b) wavenumber resonance conditions (kinematics); (c) a dynamical system deduced as a reduction of the initial PDE (dynamics).

Any combination of these approaches is very useful and provides new insights into understanding the temporal evolution of the systems under consideration. Thus, in [7], a combination of kinematic results and a numerical study of the initial PDE allows us to establish through which modes the energy flows into the wave field from the exact resonant triad. The combination of kinematics and dynamic description of large resonant clusters of complex structure makes possible the use of stochastic methods of analysis for describing the temporal dynamics of a cluster [23]. Using the form of a dynamic system for resonantly interacting waves to transform the original nonlinear PDE allows you to convert it into a more convenient form for further research, and in some cases, linearize the original equation [24].

The main conclusions of our study can be formulated as follows.

*1      Detuning cannot be arbitrary small.*

The study of detuned resonances at the level of kinematics operates with the notion of characteristic detuning, which may make the results obtained by this approach nonphysical. Before studying detuned resonances in a specific problem, one has to compute the set of its physical detunings according to the set of eigenfunctions of the initial PDE. The set of physical detunings is discrete and finite in a fixed spectral domain. Other choices of detuning correspond to changing the initial energy-conserving PDE to a different PDE with additional forcing or dissipation terms included.

*2      Detuned and exact resonances are not necessarily close in the Fourier space.*

The intuitive belief that detuned resonances should be studied in a small neighborhood of an exact resonance is erroneous: the detuned resonances with a very small detuning can exist far away in the spectral domain from the exact resonance. However, there may be a special choice of initial energy distribution in a wave system, e.g., as in [8], for which the study of detuned resonances in the vicinity of exact resonances is appropriate.

*3      Zero-frequency modes participate in detuned resonances.*

The study of detuned resonances should take into account the existence of zero-frequency modes which are of no importance while studying three-wave exact resonances. As can be seen from Figures 6 and 7, zero-frequency modes not only have their own complex dynamics, but also change the dynamics of the exact resonant triad, being coupled with it.

As already said, in this paper, we aim to lay down the common language and foundations for a subsequent study of detuned resonances. Our choice of examples is in no way exhaustive or complete but it serves its purpose: it draws the attention of the researcher to three main mistakes in the study of detuned resonances which should be avoided.

**Author Contributions:** Conceptualization, E.T.; methodology, E.T. and G.C.; software, G.C. and Y.A.; validation, G.C., Y.A. and E.T.; writing—original draft preparation, G.C.; writing—review and editing, E.T. All authors have read and agreed to the published version of the manuscript.

**Funding:** Open Access Funding by the Austrian Science Fund (FWF) under projects P30887 and P31163.

**Institutional Review Board Statement:** Not applicable.

**Informed Consent Statement:** Not applicable.

**Acknowledgments:** This research was initiated at the 9th Festival de Theorie in Aix-en-Provence; we wish to thank the organizers for their facilitation and for financial assistance. GC also thanks the Johannes Kepler University Supplementary for its hospitality during a visit to Linz, and acknowledges the financial support of the Leverhulme Trust. GC also thank G. Vallis, S. Thomson, B. Wingate, and A. Owen for helpful discussions.

**Conflicts of Interest:** The authors declare no conflict of interest. The funders had no role in the design of the study; in the collection, analyses, or interpretation of data; in the writing of the manuscript; or in the decision to publish the results.

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
