# Peer review of "Detuned Resonances"

_fluids, doi:10.3390/fluids7090297_

Round 1
Reviewer 1 Report
The manuscript aims to establish a common language and foundations for a subsequent study of detuned resonances which has remained an ambiguous concept despite of demonstrating its significance in diverse research areas such as fluid dynamics or geophysics. The corresponding author of the current manuscript has also co-published a significant work on detuned resonances in her 2020 Mathematics (MDPI) paper where it was established that by suitably chosing the detuning values one can extend the energy variation range beyond the range of the exact resonance. According to the authors of the submitted maniscript, this is the very first time an effort was made to "introduce some systematic terminology to the field and give illustrative examples confirming the importance of understanding what resonance detuning is." I completely agree with this claim. I think this article will provide new insights into understanding the significance of detuned resonances applied to the systems originated from diverse physical phenomena.
Author Response
Thank you very much for the review
Reviewer 2 Report
The detuned resonances can be even higher than in tuned cases, although the strong base flow is non-self-similarity. How do the authors explain this unusual phenomenon?
Author Response
>The detuned resonances can be even higher than in tuned cases, although the >strong base flow is non-self-similarity. How do the authors explain this unusual >phenomenon?
The detuning may be physical (that is, corresponding to the original PDE and its eigenfunctions) and artificial (arbitrary chosen while studying dynamical system). Accordingly, the detuned resonances can be higher than in tuned case when detuning is artificial and can be made physical only if the original PDE is modified by adding an external force.
Reviewer 3 Report
This contribution approaches the interesting topic of detuned resonances which is a topic of widespread multidisciplinary interest describing many physical, mechanical, biological and other systems. The paper establishes the fundamentals of detuned resonances and pinpoints three main conclusions: 1. Detuning cannot be arbitrarily small, 2. Detuned and exact resonances are not necessarily close in the Fourier space and 3. Zero-frequency modes participate in detuned resonances.
I find the paper interesting and very appropriate for the journal. Also English is very good and I support publication basically in its actual form. Nevertheless, two specific comments are that the format of the paper should be accommodated to the specific template of the journal and the affiliations of the authors must be completed
Author Response
>format of the paper should be accommodated to the specific template of the >journal and the affiliations of the authors must be completed
this will be done